# Radioimaging in the Evaluation of the Therapeutic Effect of the Vegetable Extract Obtained from Epilobium Parviflorum Schreb

Erdogan Elvis Șachir [1,*,†], Cristina Gabriela Pușcașu [1,*,†], Aureliana Caraiane [1,†], Gheorghe Raftu [1,†], Victoria Badea [1,†], Cristina Bartok-Nicolae [1,*,†], Carmen Grierosu [2,†] and Ramona Feier [3,†]

[1] Department of Dentistry, Faculty of Dentistry, "Ovidius" University of Constanța, 7 Ilarie Voronca Street, 900684 Constanța, Romania; drcaraiane@yahoo.com (A.C.); gheorgheraftu@yahoo.com (G.R.); badea_victoria@yahoo.com (V.B.)

[2] Faculty of Dentistry, Apollonia University of Iasi, Păcurari 11, 700511 Iași, Romania; grierosucarmen@yahoo.com

[3] Department of Dentistry, Faculty of Dentistry, Dimitrie Cantemir of Tirgu Mures, 3-5 Bodoni Sandor Street, 540545 Tirgu Mureș, Romania; dr.ramonafeier@yahoo.ro

* Correspondence: erdogan_sachir@yahoo.com (E.E.Ș.); cristinap@gmb.ro (C.G.P.); dr.cristina_nicolae@yahoo.com (C.B.-N.); Tel.: +40-745074720 (C.B.-N.)

† All authors contributed equally to this work.

**Abstract:** For years, apical microleakage has been considered the main factor in endodontic failure therapy. Sealing abilities and antibacterial properties of root canal sealers and intracanal medicaments between appointments have been recognized as important factors for the success of endodontic treatment. Background: Apical periodontitis (AP) is an inflammatory disease around the apex of a tooth root. The microorganisms reach the pulp by dentinal tubules especially when there is an open cavity after a coronal fracture and the pulp is in contact with the septic oral environment. The aim of the study was to evaluate the dynamics of healing by recording periapical index (PAI), after two appointment endodontic procedure with commercial or experimental intracanal medicament. Methods: A total of 40 patients with primary chronic apical periodontitis requiring root canal treatment were assigned randomly into four groups according to the teeth medicated with dehydrated plant extract, calcium hydroxide, calcium hydroxide mixed with chlorhexidine (CHX) gel 2%, Walkhoff paste and obturated on a second visit, 7 days later. Patients were recalled at intervals of 3, 6, and 12 months to evaluate the treated teeth both clinically and radiographically for periapical healing. A 5-score scale PAI was used to evaluate stages of the periapical healing on a periapical radiography using a Kodak Dental imaging software provided by the radio-imagistic center. Results: Radiological evaluation revealed that the experimental intracanal medicament had a cumulative positive healing capacity by reducing the PAI as well as all resorbable pastes used in endodontic conventional therapy. Conclusions: The results suggest that the vegetable dry extract obtained from *Epilobium parviflorum Schreb* can be used as an inter-appointment medication among with the root canal filling for the positive effect on apical healing quantified by reducing the PAI.

**Keywords:** dry vegetable extract; periapical index score; endodontic treatment

## 1. Introduction

The infection of the dental root canal system is called an endodontic infection [1]. Most endodontic infections are caused by bacteriawhich can cause an apical periodontitis [2]. This is viewed as a dynamic encounter between microbial factors and host defenses at the interface between infected radicular pulp and periodontal ligament that results in local inflammation, resorption of hard tissues, destruction of other periapical tissues, and eventual formation of various histopathological categories of apical periodontitis [3].The microorganisms reach the pulp by dentinal tubules especially when there is an open cavity after a coronal fracture and the pulp is in contact with the septic oral environment. Another

pathway for the microorganism is the periodontal membrane when they use the lateral channel or the apical foramen [1]. Studies have shown that apical periodontitis could be associated with endodontic treatments. Despite technological advances made during the last couple of decades, studies continue to find high frequencies of substandard root fillings. Some reports show improvement in the quality of root fillings as assessed radiographically, but without a concomitant improvement in the apical health in connection with the root-filled teeth [4].Various studies have elucidated the importance of quality of root canal filling and coronal seal in the success for non-surgical endodontic therapy [5].

The imagistic diagnostic techniques are essential from the diagnostic viewpoint in various domains of the dental medicine. As the aim of the endodontic treatment is to maintain healthy periapical status, the radiographic exam allows the assessment of the changes of the periapical area related to the bone density and the progression of the periapical inflammation [6]. The absence of the periapical radio transparence confirms the effectiveness of the cleaning, disinfection and sealing provided by the root filling and coronal filling [7]. Ideal features of an imaging system are geometric accuracy, minimal superimposition, ease of availability and usage, reliable, reproducible, relatively inexpensive, and most importantly minimum radiation exposure to the patient [8]. Superimposition and image distortion can be some potential disadvantages of conventional radiographic examination which increased the interest for cone beam computed tomography (CBCT), despite the limited use of only 17% investigations in the endodontics field. However, CBCT should not be used routinely in the diagnosis of periapical lesions and endodontic applications due to the ALARA (As Low As Reasonably Achievable) [6,8].

Radiographic evaluation of the apical periodontitis can be made using the PAI scoring system given by Ørstavik in 1986. This is a five-point scale radiographic interpretation aimed to assess the absence, presence or changing of a diseased state. The reference is made up of a set of five radiographs with corresponding line drawings and their associated score on a photographic print [9]. Table 1 represents the description of PAI scores.

**Table 1.** Description of Periapical index scores (adapted from Ørstavik et al. [9]).

| PAI Score | Description of Radiographic Findings |
| --- | --- |
| 1 | Normal Periapical Structures |
| 2 | Small changes in Bone Structures |
| 3 | Change in Bone Structure with Mineral Loss |
| 4 | Periodontitis with well-defined radiolucent area |
| 5 | Severe periodontitis with exacerbating features |

Endodontic materials are used in endodontic treatment, which is the procedure to save the tooth when the pulp and/or peri radicular tissues are injured. These materials can be generally classified into two groups: materials used to maintain the vitality of the pulp (pulp capping materials) and materials used to disinfect (irrigants and intracanal medicaments) and fill the pulp in root canal therapy [10].

The use of an interappointment medicament has been shown to significantly improve disinfection after chemo mechanical procedures [11]. Calcium hydroxide is a commonly used intracanal medicament. It possesses several advantages such as tissue dissolving ability and antibacterial properties [12]. Chlorhexidine (CHX) is an antiseptic with a broad antimicrobial spectrum and high substantivity. It is commonly used as an interappointment root canal medication [13].

Camphor mono-chlorophenol (CMCP) has high antimicrobial activity. However, the use of formocresol is controversial due to its potential cytotoxicity. It has been associated with carcinogenicity, immunological changes, cytotoxicity, teratogenicity, mutagenic effects and causing enamel defects in permanent teeth and systemic changes in internal organs such as the kidneys and the liver [14]. CMCP is also a phenolic derivative that can stimulate periapical tissues at higher concentrations. Therefore, an alternative material with high efficacies would be of utmost clinical interest [15].

Recently, the focus of research has shifted towards finding natural alternatives to synthetic medications [16]. Several natural products have been tested as intracanal medicaments for this purpose, namely: Liquorice extract, propolis, Morinda controflia, and the extract of Arctium lappa [17]. Propolis is a natural, biocompatible, and resinous byproduct, which has been used extensively as an organic medicine for managing oral and throat infection, dental caries [18]. It possesses anti-microbial and anti-inflammatory properties due to the presence of flavonoids. Moreover, the anti-microbial properties of propolis have been proven to be superior to calcium hydroxide in vitro [19,20]. Similarly, two clinical studies (Parolia A. et al., 2010, Jolly M. et al., 2013) have found propolis to be superior to calcium hydroxide when used as a direct pulp capping agent and as an intracanal irrigant [21,22].

The anti-inflammatory and antiproliferative activity of *Epilobium parviflorum Schreb* extracts was initially studied on cells in the context of benign prostatic hyperplasia, and in parallel the analgesic, antioxidant, antibacterial and antifungal action were studied [23–25]. Plant polyphenols represent a group of chemical substances ubiquitously distributed in all higher plants. These secondary metabolites possess free radical scavenging and antimicrobial activity. These properties can be advantageously exploited, especially because of the abundance of polyphenols and their derivatives in various agricultural and food industry waste and by-products and the possibility of convenient extraction by either organic or aqueous solvents [26–28].

There are studies (V. Steenkamp et al., 2006, Bajer T. et al., 2017) regarding the antibacterial activity of *Epilobium parviflorum Schreb* species but none of them assessed the efficacy of *Epilobium parviflorum Schreb* as a "medicament" for root canal treatment in apical periodontitis; therefore, this study aimed to evaluate the dynamics of healing by recording PAI after two appointment endodontic procedure with commercial or experimental intracanal medicament between the appointments [29,30].

## 2. Materials and Methods

### 2.1. Dry Vegetable Extract Acquisition Process

Plants were picked during the summer period from rural region of Dobrogea area and left to dry for one month. In the technological process, a drying yield of 20% was used.

In a borosilicate container, 50 g of dried *Epilobium parviflorum Schreb* was added to 100 mL of 95% ethanol and 100 mL distilled water, then stored at room temperature for 7 days. The technique used was cold maceration followed by vacuum pump filtration with a pore membrane diameter 0.22 μm, to obtain a load free of germ. In order to be placed as a root canal paste, the hydroalcoholic solution of *Epilobium parviflorum Schreb* was dehydrated by means of a rotary evaporator with a vertical condenser obtaining a dry vegetable extract with a concentration of 10 mg/mL.

### 2.2. Study Group

A prospective study was conducted in Constanța, on 60 adults, who required endodontic therapy, in the Endodontics Department of Dentistry Faculty, "Ovidius" University of Constanța. Ethical approval for the study was obtained from the Bioethics committee of Ovidius University 15547/09.11.2018. The study was conducted following the Helsinki declaration revised in 2013, and evaluation took place over a period of two years, starting on December 2018 and finishing on November 2020. All patients with posterior teeth diagnosed with pulp necrosis, (i.e., negative response to pulp sensitivity test with cold stimulus, confirmed by absence of bleeding during access cavity preparation) asymptomatic or symptomatic apical periodontitis were considered eligible and included in the study group after signed informed consent for the research. All personal data collected from the subject were blinded and stored with the given unique registration number which was used for random distribution in one of the study groups, and the sequence of distribution was 4. Based on the intracanal medication commercial (calcium hydroxide, calcium hydroxide mixed with CHX gel 2%, Walkhoff paste) or experimental (dehydrated plant extract of 10 mg/mL concentration) the study group was divided into 4 groups of 10 patients each.

The measurements obtained were statistically analyzed using SPSS software, version 25.0 (IBM Inc., Chicago, IL, USA).

The inclusion criteria in the study were: participants had to be over 18 years of age without allergic reactions with posterior teeth diagnosed with pulp necrosis. Moreover, the included teeth also had symptomatic apical periodontitis, asymptomatic apical periodontitis, or chronic apical abscess associated with a periapical radiolucency; absence of root fracture of the involved tooth, absence of periodontal pocket. The exclusion criteria from the study were: patients who are unable to appear for periodic check-ups, teeth with simple decay, teeth with hyperemia, poor oral hygiene, absence of enough tooth structure for rubber dam isolation, patients with contributory medical history, patients who received antibiotic therapy during the previous 6 month.

At the beginning of the study, 60 patients with asymptomatic or symptomatic apical periodontitis were examined; only 40 of them were accepted for the inclusion in study based on acceptance criteria.

Root canal treatment was performed in accordance with the European Society of Endodontology guidelines (European Society of Endodontology 2006) [31].

*2.3. Endodontic Treatment Protocol*

After clinical examination patients were referred to a digital radio imaging center to assess the initial status by a periapical radiography. All patients in the study had a standardized X-ray series. Preoperative, immediate postoperative, and recall radiographs were taken with individual bite blocks attached to the beam guiding device, Rinn Xcp holder (Dentsply Maillefer, Switzerland) [32]. All radiographic sensors were exposed and processed digitally under similar conditions.

Root treatment procedures were completed under the same protocols by an experienced endodontist. All treatments were performed under local anesthesia (Lidocaine spray 10%, Mepivastesin 3%/Ubistesin 4% 3M ESPE Germany). Single tooth isolation was undertaken using a rubber dam and light-cured gingival barrier (Dam Liquid, Cerkamed, Stalowa Wola, Poland). The surface of the rubber dam and the tooth were disinfected by swabbing for 60 s with rubbing alcohol 70% (ethanol-based liquid, Alcomar, Romania). Coronal access was achieved with a sterile bur followed by canals initial scouting with size 10 or size 15 Kerr file (Dentsply Sirona). Working length was determined using an electronic apex locator (Denjoy Joypex 5, Skysea) and confirmed with a digital periapical radiograph. Canal preparation was performed in a crown-down approach using rotary instruments (ProTaper$^{®}$ Gold; Dentsply Sirona) at 300 RPM and a maximum torque of 4N to a size F2 master apical size, whenever possible [33].

For the teeth with previously root canal treatment, former root canal obturation material was mechanically removed using ProTaper$^{®}$ RT files (Dentsply Maillefer, Ballaigues, Switzerland) coded D1 to D3. After reaching the original apical foramen using hand files from #6-15; working length was determined and the root instrumentation performed as per the above-described technique [34].

Inthe commercial medicament groups, the canals were disinfected with 2% sodium hypochlorite (NaOCl) throughout the procedure delivered using a side-vented 30-gauge needle (Mekan Med., Shanghai, China). The NaOCl activations were made by ultrasonic passive technique using the protocol described by Pirela et al. (2020) [35]. Final irrigation was made with 2% NaOCl followed by sterile saline solution, and 17% ethylenediaminetetraacetic acid (EDTA, Endo-solution, Cerkamed, Stalowa Wola, Poland). After a final rinse with saline, the root canals were dried with sterile paper points (Absorbent Paper Points, Metabiomed). Only the NaOCl was entirely replaced with the hydroalcoholic plant extract 25% on the group with the experimental medicament, the rest of the irrigation protocol remain unchanged.

An interappointment dressing of experimental or commercial paste was placed repetitively with a Kerr file 20 ISO by counterclockwise rotational movements associated with

vertical plunging movements until the intracanal medication suppressed the cavity access; a temporarily dressed glass ionomer cement (GC Fuji, MN, USA) was used [36].

### 2.4. Root Canal Filling

After 1 week, the root canals were re-entered; in the groups with commercial medicament, the temporary antiseptic paste from the root canal was removed using a copious irrigation with NaOCl 2%, EDTA 17%, citric acid 40% (Cerkamed, Stalowa Wola, Poland) in combination with hand filing (Kerr file 25 ISO). For more efficient elimination of intracanal medicament, passive ultrasonics were used according tothe protocol described by Raghu et al. (2017). This is accomplished with a small file (20 or 25 ISO) vibrated in a previously shaped root canal to produce acoustic streaming that transfers its energy to the irrigant inside the canal. In the group with the experimental medicament, the above protocol of removal temporary paste was used, only the NaOCl was entirely replaced with the hydroalcoholic plant extract 25% [37].

Filling of the root canals was performed by continuous wave of condensation technique (System B; Kerr Endo, Orange, CA, USA) and thermoplasticized gutta-percha injection (Obtura II, Spartan, IL, USA). Coronal parts of root canal systems were sealed using glass-ionomer cement (Fuji IX, GC Int., Tokyo, Japan) with a minimum thickness of 1.5–2 mm. Enamel and dentin conditioning was performed with a self-etching adhesive system (GC G-BOND, GC Int., Minnetonka, MN, USA) and restored using composite material (Gradia Direct, GC Int., Minnetonka, MN, USA) placed incrementally [38,39].

After root filling, teeth were referred to a prosthetic specialist for a crown coverage if indicated. During the survey, temporary prosthetic crown according to Heimann (2021) was made by a modified radiolucent hybrid composite (Estenia, Kuraray) to avoid interference with the radiological assessment and diagnosis of marginal gaps, quality of the root filling and the overall condition for apical healing [40].

The patients were sent to the same radiological center to perform an immediate postoperative radiograph control using a periapical radiography as well as at 3, 6, 12 months recall. Radiological evaluation was performed at 3 months after endodontic treatment in accordance with other previous studies (Best S. et al., 2014, Salceanu M. et al., 2021) and with the Romanian Endodontics Guide (Andrei Iliescu, 2015) which establishes the appearance of radiologically visible apical healing signs at 3 months after performing the root canal treatment [41–43].

## 3. Results

### 3.1. Clinical Evaluation

The study group consisted of 40 patients, 22 were female (55%) and 18 were male (45%), aged between 18 up to 65 years, each of them hada single tooth with apical periodontitis to summarize a total of 40 posterior teeth (single, double or multi-rooted) treated. Clinical evaluation was performed at intervals of 3, 6 and 12 months after complete endodontic treatment revealing the absence of pain and the dental mobility where it preoperatively existed; no visible clinical signs of endodontic treatment failure exist.

### 3.2. Paraclinical Evaluation

3.2.1. Evaluation of the PAI Score on Group with Dehydrated Plant Extract

Scheme 1 shows the score of the PAI at different time intervals in the group of patients where the plant extract was applied; a linear regression of the PAI values is observed as the time interval increases, there being an inverse proportional relationship between these two parameters. Thus the longer the time interval, the lower the value of the PAI scores.

Figure 1 shows by retro-alveolar radiographs aspects of the initial periapical lesion (Figure 1a) and the healing stages after endodontic therapy at 3 months (Figure 1b), 6 months (Figure 1c), 12 months (Figure 1d) recall; there is an inverse proportional relationship between the values of the periapical index and the time interval, measured in

months from the initiation of endodontic therapy, so after long periods of time we have low values of the periapical index.

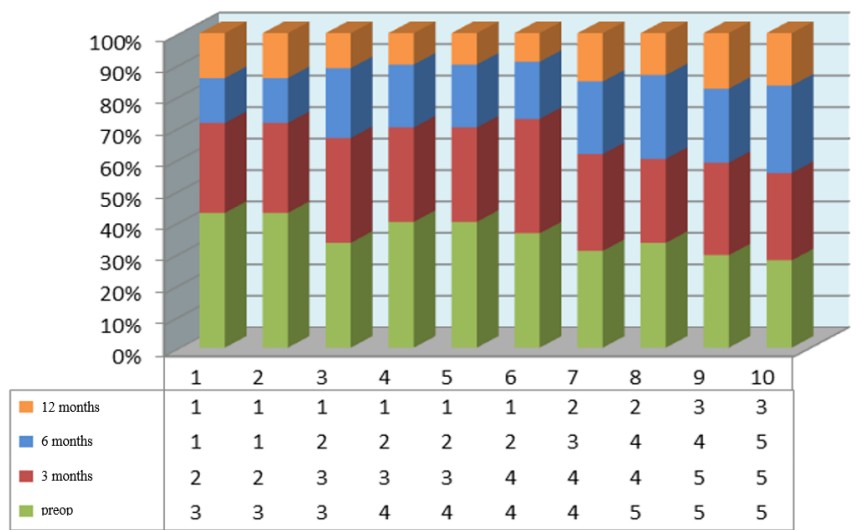

| | 1 | 2 | 3 | 4 | 5 | 6 | 7 | 8 | 9 | 10 |
|---|---|---|---|---|---|---|---|---|---|---|
| 12 months | 1 | 1 | 1 | 1 | 1 | 1 | 2 | 2 | 3 | 3 |
| 6 months | 1 | 1 | 2 | 2 | 2 | 2 | 3 | 4 | 4 | 5 |
| 3 months | 2 | 2 | 3 | 3 | 3 | 4 | 4 | 4 | 5 | 5 |
| preop | 3 | 3 | 3 | 4 | 4 | 4 | 4 | 5 | 5 | 5 |

**Scheme 1.** Distribution of the PAI over time intervals in thegroup with dehydrated plant extract.

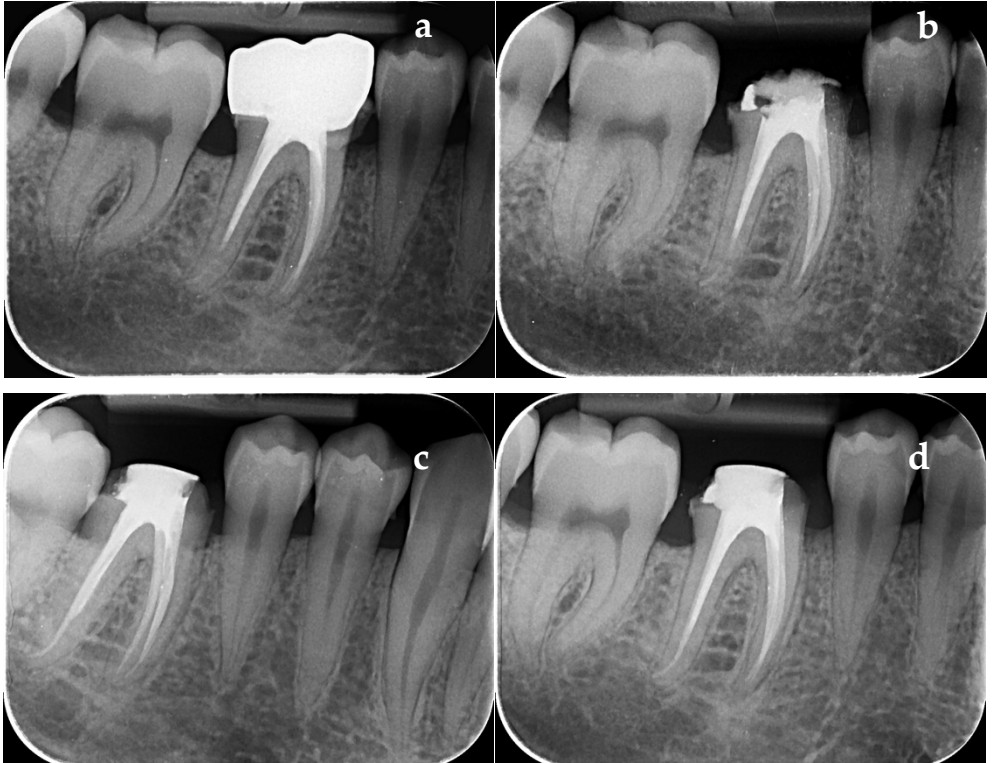

**Figure 1.** Radiological examination performed preoperatively and postoperatively at intervals of 3 months, 6 months and 12 months, (**a**) initial periapical lesion; (**b**) 3 months after endodontic treatment; (**c**) 6 months after endodontic treatment; (**d**) 12 months after endodontic treatment.

### 3.2.2. Evaluation of the PAI Score on Group with Calcium Hydroxide

Scheme 2 shows the score of the PAI at different time intervals in the group of patients where calcium hydroxide was applied; a linear regression of the PAI values is observed as the time interval increases, there being an inverse proportional relationship between these two parameters. Thus, the longer the time interval, the lower the value of the PAI scores.

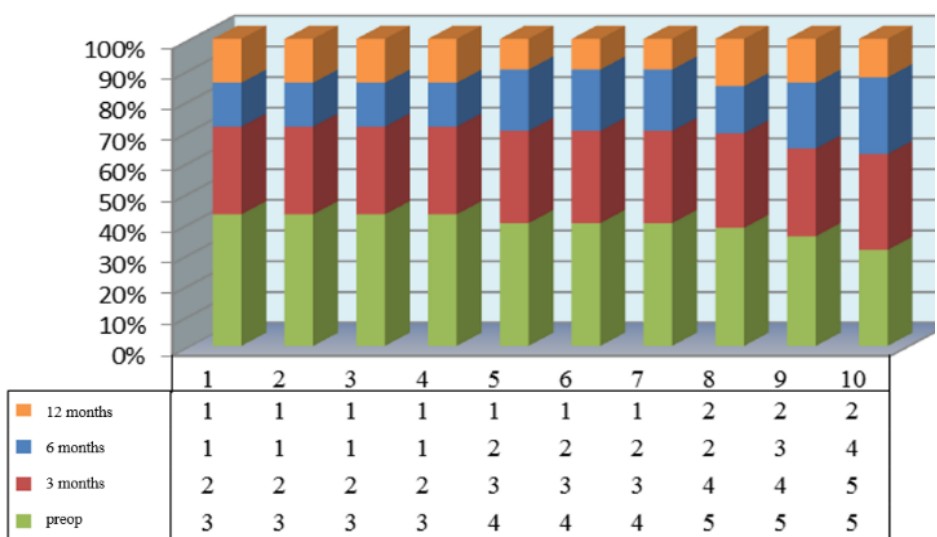

**Scheme 2.** Distribution of the PAI over time intervals in thegroup with calcium hydroxide.

### 3.2.3. Evaluation of the PAI Score on Group with Calcium Hydroxide Mixed with CHX Gel 2%

Scheme 3 shows the dimensions of the periapical lesion at different time intervals in the group of patients where calcium hydroxide was applied mixed with CHX gel 2%; a linear regression of the extent of the periapical lesion is observed as the time interval increases, there being an inverse proportional relationship between these two parameters.

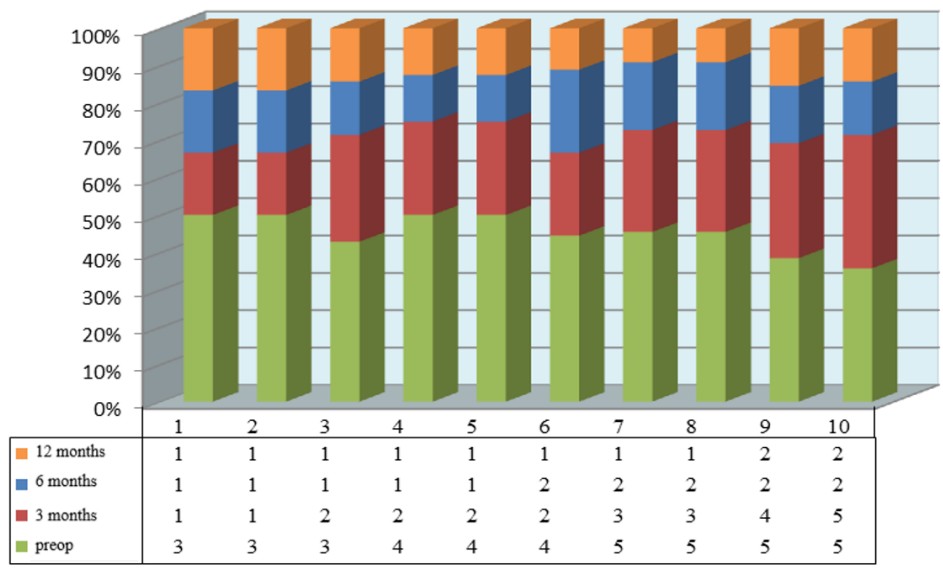

**Scheme 3.** Distribution of the PAI over time intervals in thegroup with calcium hydroxide mixed with CHX gel 2%.

### 3.2.4. Evaluation of the PAI Score on Group with Walkhoff Paste

Scheme 4 shows the dimensions of the periapical lesion at different time intervals within the group of patients where Walkhoff paste was applied; a linear regression of the extent of the periapical lesion is observed as the time interval increases, there being an inverse proportional relationship between these two parameters.

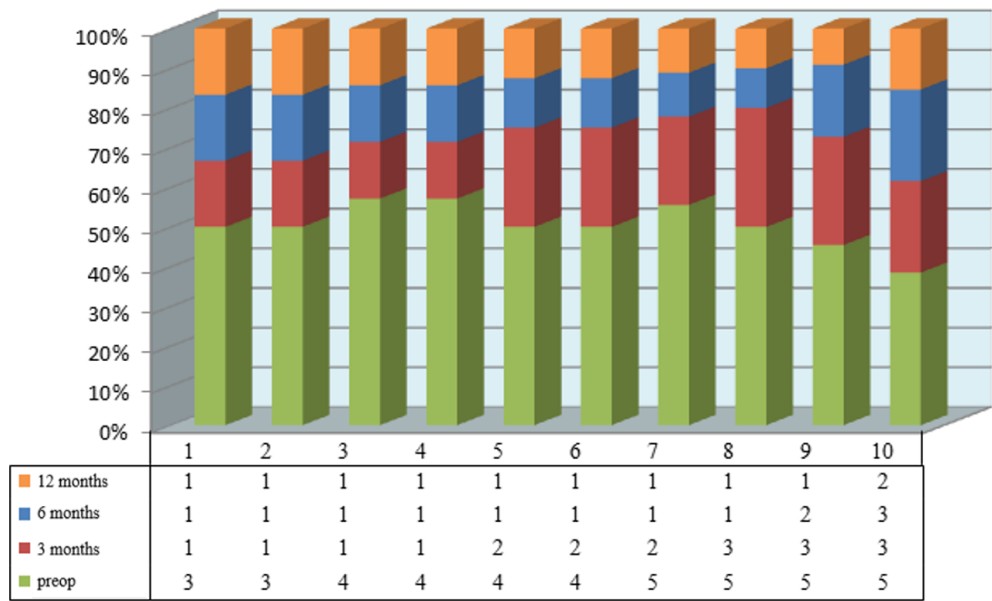

**Scheme 4.** Distribution of the PAI over time intervals in thegroup with Walkhoff paste.

Table 2 shows the PAI score at different time intervals; it has a linear regression decreasing from preop to 3, 6 and 12 months post-therapy.

**Table 2.** Mean values and standard deviation of the PAI, at different time intervals, for the group with dehydrated plant extract, calcium hydroxide, calcium hydroxide with CHX gel 2%, Walkhoff paste.

|  | Dry Vegetable Extract | $Ca(OH)_2$ | $Ca(OH)_2$ Mixed with CHX Gel 2% | Walkhoff Paste |
|---|---|---|---|---|
| **Period** | **Mean $\pm$ SD** | **Mean $\pm$ SD** | **Mean $\pm$ SD** | **Mean $\pm$ SD** |
| Preop | $4 \pm 0.73$ | $3.9 \pm 0.87$ | $4.26 \pm 0.57$ | $4.18 \pm 0.47$ |
| 3 months | $3.5 \pm 1.08$ | $3 \pm 1.05$ | $1.94 \pm 1.82$ | $1.17 \pm 1.33$ |
| 6 months | $2.6 \pm 1.34$ | $1.9 \pm 0.99$ | $0.32 \pm 0.37$ | $0.39 \pm 0.91$ |
| 12 months | $1.6 \pm 0.84$ | $1.3 \pm 0.48$ | $0.19 \pm 0.42$ | $0.03 \pm 0.11$ |

$Ca(OH)_2$ = Calcium hydroxide. SD = Standard deviation.

Table 3 shows that there is an average inverse correlation between the PAI score recorded preoperatively and after 12 months and also between the two parameters there are differences with high statistical significance ($p < 0.001$).

**Table 3.** Correlations between PAI values recorded at different time intervals.

|  | Dry Vegetable Extract | $Ca(OH)_2$ | $Ca(OH)_2$ with CHX Gel 2% | Walkhoff Paste |
|---|---|---|---|---|
| **Period** | **Correlation Index $\pm$ p** | **Correlation Index $\pm$ p** | **Correlation Index $\pm$ p** | **Correlation Index $\pm$ p** |
| preop vs. 3 months | $0.88 \pm 0.258$ | $0.95 \pm 0.001$ | $0.95 \pm 0.001$ | $0.93 \pm 0.001$ |
| preop vs. 6 months | $0.90 \pm 0.011$ | $0.84 \pm 0.001$ | $0.92 \pm 0.001$ | $0.68 \pm 0.001$ |
| preop vs. 12 months | $0.80 \pm 0.001$ | $0.80 \pm 0.001$ | $0.58 \pm 0.001$ | $0.52 \pm 0.001$ |
| 3 months vs. 6 months | $0.91 \pm 0.117$ | $0.93 \pm 0.021$ | $0.96 \pm 0.012$ | $0.72 \pm 0.146$ |
| 3 months vs. 12 months | $0.85 \pm 0.001$ | $0.87 \pm 0.001$ | $0.74 \pm 0.008$ | $0.54 \pm 0.014$ |
| 6 months vs. 12 months | $0.91 \pm 0.062$ | $0.79 \pm 0.148$ | $0.95 \pm 0.001$ | $0.91 \pm 0.231$ |

$Ca(OH)_2$ = Calcium hydroxide. SD = Standard deviation.

## 4. Discussion

The objective in this study was to assess radiographic healing of teeth treated with commercial or experimental intracanal medicament between two endodontic appointments.

To evaluate radiographic healing patterns, the PAI system was used, developed by Ørstavik et al. (1986). Since its inception, this scoring system has become increasingly popular in endodontic outcome studies [9]. At baseline, the teeth were classified as having a radiolucent lesion with a PAI score mean >4. By the latest follow-up, all of the teeth had a PAI score mean of <1.3; regression of the PAI score mean for all four substances studied demonstrates the effectiveness of the intracanal medicaments.

The ratio of cured teeth is a majority in each study group, which demonstrates the effectiveness of the substances applied in endodontic therapy alongwith the canal filling and the success of endodontic treatment by radiological monitoring of apical healing.

The results of the present study revealed significant information on intracanal medications, commercial and experimental, the role in the apical healing process monitored by recording the PAI score at 3, 6 and 12 months recall after root canal filling on a periapical radiography.

In the present study, the dehydrated vegetable extract had a therapeutic effect, managing to produce apical healing in a percentage of 60%, reducing the PAI score from 4 preoperative to 1.6 after 12 months recall. The data obtained from the statistical processing demonstrate that the therapeutic value of the experimental paste is above that of the commercial medicament used in endodontic practice. In the accessed literature we did not find studies to test the therapeutic effect of alcoholic, aqueous or dehydrated vegetable extract obtained from *Epilobium parviflorum Schreb* in endodontic therapy.

The most favorable outcome was with the Walkhoff paste which produced apical healing in a percentage of 90%, reducing the PAI score from 4.18 preoperative to 0.03 after 12 months recall; although difficult to compare due to the methods and techniques used, the data obtained in this study are consistent with those of the study by Zhila Imani et al. (2018), who demonstrated by bacteriological techniques that the therapeutic efficacy of the p-monochlorophenol solution in Walkhoff paste has a high capacity for apical healing [15].

The group with calcium hydroxide had an apical healing in a proportion of 70%, reducing the PAI score from 3.9 preoperative to 1.3 after 12 months recall. The results are similar to the data obtained by Sahara et al. (2019), who used the same method of quantifying apical healing, with the PAI decreasing from scale 5 preoperative to 2 after 3 months recall [44]. In previously published studies we found data that show that the therapeutic effect of calcium hydroxide is time-dependent as it is in the study conducted by Gheorghiu Maria et al. (2018). They have shown that calcium hydroxide has a very high efficiency when placed in the root canals for a longer period of time (10 days) and extremely low efficiency after 48 h [45].

The group with calcium hydroxide paste mixed with CHX 2% had a therapeutic effect on apical healing in a percentage of 80%, reducing the PAI score from 4.26 preoperative to 0.19 after 12 months recall. The data obtained in the present study are close to the results obtained by Ertugrul Ercan et al. (2007), who used the whole PAI at the same time intervals to evaluate periapical healing. The results obtained were 64.1%, although the CHX concentration was 1%; the difference in percentage between the two studies can be determined both by the concentration of CHX used and by its forms of presentation (liquid/gel) [46].

The limitations of the study are given by the final number of the study group and the 5-score scale PAI which is really complicated and it can lead to errors of judgment if the interpretation is made by a beginner practitioner.In order to overcome these limitations from afuture perspective, it is necessary to conduct a study with a larger number of patients and to avoid classification errors, we will use in other studies an easier PAI system in a 3-score scale described by Nardi et al. (2017) [47].

## 5. Conclusions

The results suggest that the vegetable dry extract obtained from *Epilobium parviflorum Schreb* can be used as an inter-appointment medication alongwith the root canal filling for the positive effect on apical healing quantified by reducing the PAI.

**Author Contributions:** Conceptualization, E.E.Ș., C.G.P., C.B.-N.; methodology, G.R., E.E.Ș.; validation, C.G.P., C.B.-N.; formal analysis, C.G.P., C.B.-N.; investigation, E.E.Ș.; resources, E.E.Ș., C.G., R.F.; data curation, G.R., C.B.-N.; writing—original draft preparation, E.E.Ș., C.G.P.; writing—review and editing, E.E.Ș., C.G.P., V.B., A.C., G.R., C.B.-N., C.G.P.; visualization, C.G.P., E.E.Ș., R.F., C.G., C.B.-N.; supervision V.B., A.C. All authors have read and agreed to the published version of the manuscript.

**Funding:** This research received no external funding.

**Institutional Review Board Statement:** The study was conducted according to the guidelines of the Declaration of Helsinki, and approved by the Ethics Committee of "Ovidius" University (15547/09.11.2018).

**Informed Consent Statement:** Informed consent was obtained from all subjects involved in the study.

**Data Availability Statement:** Not applicable.

**Conflicts of Interest:** The authors declare no conflict of interest.

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
