# Peer review of "Radioimaging in the Evaluation of the Therapeutic Effect of the Vegetable Extract Obtained from Epilobium Parviflorum Schreb"

_applsci, doi:10.3390/app12030998_

Round 1
Reviewer 1 Report
In my opinion the methodology/results are flawed
Abstract: The introduction can be modified
How is the technical quality of root canal filling related to the antibacterial efficacy of an agent.
The results need to be more elucidated.
The abstract in general needs further revision to further elucidate to the reader what the study is describing.
Main article:
Grammar check is required
Introduction-
Line 42: Various studies have elucidated the importance of Quality of root canal filling and coronal seal in the success for Non surgical endodontic therapy. (Example Ng et al. 2011)
Materials and methods-
Line 49: Please refrain from using I, We, Us in scientific articles
Line50-52: Sentence restructuring required
Line 76: AAE diagnostic terminology should be used
Line 83: Were all retreatment or previously initiated cases taken
Line 84: How were the patients randomised in the study groups
Was blinding done?
Line 98: The context of the paragraph with respect to this study is unclear and is written in future tense
Line 105: Other than pain what other parameters were evaluated during clinical examination of the teeth.
Endodontic treatment protocol: Basic steps should be elaborated. Rubber dam application to maintain asepsis, instrumentation technique and size of preparation, irrigant activation technique
Line 137: Technique of application of antiseptic paste can be further explained. What steps were taken to ensure complete canal fill.
Line 153: How were the respective ICM removed
Better representative case can be provided.
Results: Intergroup comparison should have been performed which required standardization of the mean pre operative PAI score for each group
Information on teeth included in the study, age and demographic details should be given
Discussion: Needs major revision
Does not elaborate on the hypothesis of the study
thanks and regards
Author Response
The authors acknowledge the useful observations and suggestions of the reviewer’s as concerns the manuscript entitled ”Radioimaging in the evaluation of the therapeutic effect of the vegetable extract obtained from Epilobium parviflorum Schreb”, co-authored by Erdogan Elvis Șachir *, Cristina Gabriela Pușcașu*, Aureliana Caraiane, Gheorghe Raftu, Victoria Badea, Cristina Bartok-Nicolae*, Grierosu Carmen and Ramona Feier.
According to the reviewer’s recommendations, all the suggestions were considered, as follows:
In my opinion the methodology/results are flawed
Abstract: The introduction can be modified
We have modified the introduction
How is the technical quality of root canal filling related to the antibacterial efficacy of an agent.
We have presented the antibacterial properties of the sealers
The results need to be more elucidated.
We have improved the result presentations
The abstract in general needs further revision to further elucidate to the reader what the study is describing.
We have revised the entire abstract
For years, apical microleakage has been considered the main factor in failure in endodontic therapy. Sealing abilities and antibacterial properties of root canal sealers and intracanal medicaments between appointments have been recognized as important factors for the success of endodontic treatment. Background: Apical periodontitis (AP) is an inflammatory disease around the apex of a tooth root. The microorganisms reach the pulp by dentinal tubules especially when there is an open cavity after a coronal fracture and the pulp is in contact with the septic oral environment. The aim was to evaluate the dynamics of healing by recording periapical index (PAI), after two appointment endodontic procedure with commercial or experimental intracanal medicament between the appointments. Methods: A total of 40 patients with primary chronic apical periodontitis requiring root canal treatment were assigned randomly into four groups according to the teeth medicated with dehydrated plant extract, calcium hydroxide, calcium hydroxide mixed with CHX gel 2%, Walkhoff paste and obturated in a second visit 7 days later. Patients were recalled at intervals of 3, 6, and 12 months to evaluate the treated teeth both clinically and radiographically for periapical healing. A 5-score scale PAI was used to evaluate stages of the periapical healing on a periapical radiography using a Kodak Dental imaging software provided by the radio-imagistic center. Results: Radiological evaluation revealed that the experimental intracanal medicament had a cumulative positive healing capacity by reducing the PAI as well as all resorbable pastes used in endodontic conventional therapy. Conclusions: The results suggest that the vegetable dry extract obtained from Epilobium parviflorum Schreb can be used as an inter-appointment medication among with the root canal filling for the apical healing quantified by reducing the PAI.
Main article:
Grammar check is required
We have checked the grammar
Introduction-
Line 42: Various studies have elucidated the importance of Quality of root canal filling and coronal seal in the success for Non surgical endodontic therapy. (Example Ng et al. 2011)
We have added the sentence in the line 42 Various studies have elucidated the importance of quality of root canal filling and coronal seal in the success for non-surgical endodontic therapy. (Example Ng et al. 2011)
Materials and methods-
Line 49: Please refrain from using I, We, Us in scientific articles
Plants were picked during the summer period from rural region of Dobrogea area and let to dry for one month.
Line50-52: Sentence restructuring required
In the technological processa drying yield of 20% was used.
Line 76: AAE diagnostic terminology should be used
We have changed with apical periodontitis - All patients diagnosed with apical periodontitis, no comorbidities or known previous allergies to any drug,were considered eligible and included in the study group after signed informed consent for the research.
Line 83: Were all retreatment or previously initiated cases taken
We had retreatment and also previously initiated taken cases
Line 84: How were the patients randomised in the study groups
All personal data collected from the subject were blinded and stored with the unique registration number given which was used for random distribution in one of the study groups, the sequence of distribution was 4. Based on the intracanal medication com-mercial (calcium hydroxide, calcium hydroxide mixed with CHX gel 2%, Walkhoff paste) or experimental (dehydrated plant extract of 10 mg / mL concentration) the study group was divided into 4 groups of 10 patients each
Was blinding done?
Yes, blinding was done assigning an unique identification number to each patients in order to protect the personal data.
Line 98: The context of the paragraph with respect to this study is unclear and is written in future tense
We have removed this paragraph from M&M section
Line 105: Other than pain what other parameters were evaluated during clinical examination of the teeth.
Extra-oral examination included palpation of masticatory, neck and shoulder muscles for comparative tenderness. Auscultation and palpation of the temporomandibular joint and assessment of the range of mandibular movement was incorporated to exclude pain originating from these structures. Clinical details about the treated tooth included:tenderness to pressure and percussion of the tooth,tenderness to palpation of adjacent soft tissues,presence of an associated sinus tract or swelling in the adjacent soft tissues,periodontal probing profile around the tooth.
Endodontic treatment protocol: Basic steps should be elaborated. Rubber dam application to maintain asepsis, instrumentation technique and size of preparation, irrigant activation technique
Root treatment procedures were completed under the same protocols by an experiencedendodontist.All treatments were performed under local anesthesia (Lidocain spray 10%, Mepivastesin 3%/Ubistesin 4% 3M ESPE Germany) and rubber dam isolation, ensuring absence of saliva leakage. After accessing the tooth,the canals were prepared by preflaring the coronal (or straight) portion prior to negotiation of the apical portion and determination of working length.The root instrumentation was performed by crown-down technique for canal negotiation and shaping. The instruments available for use included:Kerr files (Dentsply Maillefer, Ballaigues, Switzerland), rotary ProTaper instrument system (Dentsply Maillefer). The recommended minimum or optimal apical size of canal preparation was size 30.
For the teeth with previously root canal treatment, former root canal obturation material was removed mechanically using ProTaper RT files (Dentsply Maillefer, Ballaigues, Switzerland) coded D1 to D3. After reaching the original apical foramen using hand files from #6-15; working length was determined and the root instrumentation performed as the above-described technique.
On the commercial intracanal medicament groups 2 mL of sodium hypochlorite 2% (NaOCl, Chloraxid 2%, Cerkamed ) was used between each file in the root canal treatmentwith a 30-gauged endodontic irrigation needle (Mekan Med., Shanghai, China). The NaOClactivation was made by ultrasonic passive technique using the protocol described by Pirela et al. (2020);the final irrigation was performed using NaOCl 2%. On the group with the intracanal experimental medicament only the hydroalcoholic plant extract 25% was used entirely in accordance with the above-described irrigation protocol.
Line 137: Technique of application of antiseptic paste can be further explained. What steps were taken to ensure complete canal fill.
An interappointment dressing of experimental or commercial paste was placed repetitively with a Kerr file 25 ISO by counterclockwise rotational movements associated with vertical plunging movementsuntil the intracanal medication suppress in the cavity access;a temporarily dressed glass ionomer cement (GC Fuji, Minnesota, USA) was used.
Line 153: How were the respective ICM removed
After 1 week, the root canals were reentered and the temporary antiseptic paste from the root canal was removed using a Kerr file 25 ISO or a ProTaper Universal F1.
Better representative case can be provided.
We improved the case presentation by changing the radiographs
|
|
|
|
Figure 1. Radiological examination performed preoperatively and postoperatively at intervals of 3 months, 6 months and 12 months, a) Initial periapical lesion; b) 3 months after endodontic treatment; c) 6 months after endodontic treatment; d) 12 months after endodontic treatment
Results: Intergroup comparison should have been performed which required standardization of the mean pre operative PAI score for each group
We have gathered together the tables to intergroup comparison
Table 2 shows the average extent of the periapical lesion recorded at different time intervals; it has a linear regression decreasing from preop to 3-, 6- and 12- months post-therapy.
Table 2. Mean values and standard deviation of the PAI, at different time intervals, for the group of patients where the dehydrated plant extract, calcium hydroxide, calcium hydroxide with CHX gel 2%, Walkhoff paste were applied.
|
|
Dry vegetable extract |
Ca(OH)2 |
Ca(OH)2 with CHX gel 2% |
Walkhoff paste |
|
Period |
Mean ± SD |
Mean ± SD |
Mean ± SD |
Mean ± SD |
|
Preop |
4±0,73 |
3,9±0,87 |
4,26±0,57 |
4,18±0,47 |
|
3 months |
3,5±1,08 |
3±1,05 |
1,94±1,82 |
1,17±1,33 |
|
6 months |
2,6±1,34 |
1,9±0,99 |
0,32±0,37 |
0,39±0,91 |
|
12 months |
1,6±0,84 |
1,3±0,48 |
0,19±0,42 |
0,03±0,11 |
Ca(OH)2=Calcium hydroxide
SD=Standard deviation
Table 3 shows that there is an average inverse correlation between the value of periapical lesion extent recorded preoperatively and after 12 monthsand also between the two parameters there are differences with high statistical significance (p <0.001).
Table 3. Correlations between PAI values recorded at different time intervals.
|
|
Dry vegetable extract |
Ca(OH)2 |
Ca(OH)2 with CHX gel 2% |
Walkhoff paste |
|
Period |
Correlation index ± p |
Correlation index ± p |
Correlation index ± p |
Correlation index ± p |
|
preop vs. 3 months |
0,88±0,258 |
0,95±0,001 |
0,95±0,001 |
0,93±0,001 |
|
preop vs. 6 months |
0,90±0,011 |
0,84±0,001 |
0,92±0,001 |
0,68±0,001 |
|
preop vs. 12 months |
0,80±0,001 |
0,80±0,001 |
0,58±0,001 |
0,52±0,001 |
|
3 months vs. 6 months |
0,91±0,117 |
0,93±0,021 |
0,96±0,012 |
0,72±0,146 |
|
3 months vs. 12 months |
0,85±0,001 |
0,87±0,001 |
0,74±0,008 |
0,54±0,014 |
|
6 months vs. 12 months |
0,91±0,062 |
0,79±0,148 |
0,95±0,001 |
0,91±0,231 |
Ca(OH)2=Calcium hydroxide
SD=Standard deviation
Information on teeth included in the study, age and demographic details should be given
The study group consisted of 40 patients of whom 22 were female (55%) and 18 were male (45%), aged between 18 up to 65 years, each of them has a single tooth with apical periodontitis summarize a total of 40 teeth (single, double or multi-rooted) treated.
Discussion: Needs major revision
Does not elaborate on the hypothesis of the study
The limitations of the study are given by the final number of the study group and the 5-score scale PAI which is really complicated and it can lead to errors of judgment if the interpretation is made by a beginner practitioner, in order to overcame these limitations for future perspective, it is necessary to conduct a study with a larger number of patients and to avoid classification errors we will use in other studies an easier PAI system in a 3-score scale described by Nardi et al. (2017) [24].
In the present study the paste obtained from the dehydrated vegetable extract had a therapeutic effect, managing to produce apical healing in a percentage of 60%, reducing the PAI score from 4 preoperative to 1,6 after 12 months recall. The data obtained from the statistical processing demonstrate, that the therapeutic value of the experimental paste is above to that of the commercial medicament used in endodontic practice. In the accessed literature we did not find studies to test the therapeutic effect of alcoholic, aqueous or dehydrated vegetable extract obtained from Epilobium parviflorum Schreb in endodontic therapy.
The most favorable outcome was with the Walkhoff paste which produced apical healing in a percentage of 90%, reducing the PAI score from 4,18 preoperative to 0,03 after 12 months recall; although difficult to compare due to the methods and techniques used, the data obtained in this study are consistent with those of the study by Zhila Imani et al (2018), who demonstrated by bacteriological techniques the therapeutic efficacy of the p-monochlorophenol solution in Walkhoff paste a high capacity for apical healing [11].
The group with calcium hydroxide had an apical healing in a proportion of 70%, reducing the PAI score from 3,9 preoperative to 1,3 after 12 months recall. The results are similar to the data obtained by Sahara et al. (2019) which using the same method of quantifying apical healing, PAI decreased from scale 5 preoperative to 2 after 3 months recall [25]. In previously published studies we found data that show that the therapeutic effect of calcium hydroxide is time dependent as it is in the study conducted by Gheorghiu Maria et al. (2018). They have shown that calcium hydroxide has a very high efficiency when placed in the root canals for a longer period of time (10 days) and extremely low efficiency after 48 hours [26].
The group with calcium hydroxide paste mixed with CHX 2% had a therapeutic effect on apical healing in a percentage of 80%, reducing the PAI score from 4,26 preoperative to 0,19 after 12 months recall. The data obtained in the present study are close to the results obtained by Ertugrul Ercan et al. (2007), who used the whole PAI at the same time intervals to evaluate periapical healing. The results obtained were 64.1%, although the CHX concentration was 1%; the difference in percentage between the two studies can be determined both by the concentration of CHX used and by its forms of presentation (liquid/gel) [27].
The results of the present study revealed significant information on intracanal medications, commercial and experimental, role in the apical healing process monitored by recording the PAI score at 3, 6 and 12 months recall after root canal filling on a periapical radiography.
We are very grateful to you for the review report and for the extremely useful suggestions!
Sincerely,
Dr. Erdogan Elvis Șachir
Reviewer 2 Report
Abstract
Ok
Key words
- OK
Introduction
- Should you better explain the role of periapical radiography, panoramic radiography, and cone beam CT in the identification of apical periodontitis. Are there also differences between the above-mentioned techniques in cases of root canal treated and root canal non-treated?
Materials and methods
Study group
- When did start and finish your study?
- Why did you choose a minimum size of 2 mm for periapical radioluncencies?
- Why did you choose a size of 4 mm for periodontal pockets?
- How did you define an incomplete root filling?
- How did you define failed endodontic treatments or chronic periodontitis?
- “Once eligibility was confirmed and after written and verbal informed consent was obtained, the patient was randomly assigned to one of following four groups according to the intracanal medicament used: dehydrated plant extract, calcium hydroxide, calcium hydroxide mixed with CHX gel 2%, Walkhoff paste.” You wrote it in a more detailed method in paragraph 2.5. Please, remove this sentence from the paragraph 2.2. It is a repetition.
Treatment planning
- What does “Treatment should be planned for those teeth that are functionally or aesthetically 98 important and have reasonable prognosis” mean?
Clinical examination
- What do you mean for “After clinical examination, patients were referred to a digital imaging center for 110 paraclinical examination”?
Recording the PAI score
- What kind of imaging technique did you use?... periapical radiography, panoramic radiography?
- I cannot understand the following sentence “The reference template is made by making a set of five radiographs together with the line corresponding to the drawings and their associated score on a 118 printed photograph.”
Endodontic treatment protocol
- What does “After the initial radiography on the teeth that presented prosthetic crowns were sectioned and removed to enhance the endodontic treatment” mean?
- What is PMMA?
Root canal filling
- What kind of imaging technique did you use? “The patients were sent to the same radiological center to perform a radiograph control the same day after root canal filling, as well as at 3, 6, 12 months control”.
Results
Clinical evaluation
- How did you decide the different stages of apical periodontitis?
- Generally, make results easier to understand. Some concepts are repeated more times?
Discussion
- Describe the limitations of your study. In addition, how could be the limitations overcome?
- Please, in the discussion section add the possibility to use a modified PAI system as described by Nardi et al. in his papers on apical periodontitis. Evaluate PAI in a 5-score scale is really complicated. It can lead to errors of judgment. For future perspective is advisable to use an easier PAI system in a 3-score scale.
Conclusions
- Ok
Figures
- Ok
Tables
- Gather together Tables 1-3-5-7
- Gather together Tables 2-4-6-8
Author Response
The authors acknowledge the useful observations and suggestions of the reviewer’s as concerns the manuscript entitled Radioimaging in the evaluation of the therapeutic effect of the vegetable extract obtained from Epilobium parviflorum Schreb, co-authored by Erdogan Elvis Șachir *, Cristina Gabriela Pușcașu*, Aureliana Caraiane, Gheorghe Raftu, Victoria Badea, Cristina Bartok-Nicolae*, Grierosu Carmen and Ramona Feier.
According to the reviewer’s recommendations, all the suggestions were considered, as follows:
Abstract
Ok
Key words
-OK
Introduction
- Should you better explain the role of periapical radiography, panoramic radiography, and cone beam CT in the identification of apical periodontitis. Are there also differences between the above-mentioned techniques in cases of root canal treated and root canal non-treated?
The infection of the dental root canal system is called an endodontic infection. This infection can cause an apical periodontitis and the progression of different forms of apical periodontitis is due to some microorganisms [1–3]. The microorganisms reach the pulp by dentinal tubules especially when there is an open cavity after a coronal fracture and the pulp is in contact with the septic oral environment. Another pathway for the microorganism is the periodontal membrane when they use the lateral channel or the apical foramen [4].
The imagistic diagnostic techniques are essential from the diagnostic in various domains of the dental medicine. As the aim of the endodontic treatment is to maintain healthy periapical status, the radiographic exam allows the assessment of the changes of the periapical area related to the bone density and the progression of the periapical inflammation. The absence of the periapical radio transparence confirms the effectiveness of the cleaning, disinfection and sealing provided by the root filling and coronal filling [4]. The conventional radiography provides to the practitioner an acceptable, effective and cheap method. The limitation of conventional radiographic exam increased the interest for cone beam computed tomography (CBCT), Despite the limited use of only 17% investigations in endodontics field [5]. However, CBCT should not be used routinely in the diagnosis of periapical lesions and endodontic applications due to the ALARA (As Low As Reasonably Achievable) principles.
Materials and methods
Study group
- When did start and finish your study?
started on December 2018 and finished on November 2020
- Why did you choose a minimum size of 2 mm for periapical radioluncencies?
We dropped on this criterion
- Why did you choose a size of 4 mm for periodontal pockets?
We dropped on this criterion
- How did you define an incomplete root filling?
We dropped on this criterion
- How did you define failed endodontic treatments or chronic periodontitis?
Endodontic failures due to poor quality obturations which are underextended oron the other way by some canals which were left untreated. We dropped on chronic periodontitis it is redundant.
“Once eligibility was confirmed and after written and verbal informed consent was obtained, the patient was randomly assigned to one of following four groups according to the intracanal medicament used: dehydrated plant extract, calcium hydroxide, calcium hydroxide mixed with CHX gel 2%, Walkhoff paste.” You wrote it in a more detailed method in paragraph 2.5. Please, remove this sentence from the paragraph 2.2. It is a repetition.
We removed it.
Treatment planning
- What does “Treatment should be planned for those teeth that are functionally or aesthetically 98 important and have reasonable prognosis” mean?
We dropped on this phrase
Clinical examination
- What do you mean for “After clinical examination, patients were referred to a digital imaging center for 110 paraclinical examination”?
After clinical examination was performed, patients were referred to a digital radio imaging center to asses the initial status by a periapical radiography. All patients in the study had a standardized X-ray series. Preoperative, immediate postoperative, and recall radiographs were taken with individual bite blocks attached to the beam guiding device, Rinn Xcp holder (Dentsply Maillefer, Switzerland). All radiographic sensors were exposed and processed digitaly under similar conditions.
Recording the PAI score
- What kind of imaging technique did you use?... periapical radiography, panoramic radiography?
Kodak Dental imaging software provided by the radio imaging center was used to perform the measurements on a periapical radiography.
- I cannot understand the following sentence “The reference template is made by making a set of five radiographs together with the line corresponding to the drawings and their associated score on a 118 printed photograph.”
Radiographic evaluation was done using the PAI scoring system given by Orstavik in 1986. This is a 5-point scale radiographic interpretation designed to determine the absence, presence, or transformation of a diseased state.The reference is made up of a set of five radiographs with corresponding line drawings and their associated score on a photographic print. Table 1 represents the description of PAI scores.
Table 1. Description of Periapical index scores (adapted from Orstavik et al. [19])
|
PAI Score |
Description of Radiographic findings |
|
1 |
Normal Periapical Structures |
|
2 |
Small changes in Bone Structures |
|
3 |
Change in Bone Structure with Mineral Loss |
|
4 |
Periodontitis with well-defined radiolucent area |
|
5 |
Severe periodontitis with exacerbating features |
Endodontic treatment protocol
- What does “After the initial radiography on the teeth that presented prosthetic crowns were sectioned and removed to enhance the endodontic treatment” mean?
We dropped on this phrase (147-150)
- What is PMMA?
We dropped on this phrase (147-150)
Root canal filling
- What kind of imaging technique did you use?
The patients were sent to the same radiological center to perform a radiograph control using a periapical radiography on the same day after root canal filling, as well as at 3, 6, 12 months recall.
Results
Clinical evaluation
- How did you decide the different stages of apical periodontitis?
Different stages of apical periodontitis were evaluated by an experimented endodontist based on periapical radiography using thePAI.
Discussion
- Describe the limitations of your study. In addition, how could be the limitations overcome?
The limitations of the study are given by the final number of the study group, in order to overcame these limitations in the future, it is necessary to conduct a study with a larger number of patients.
- Please, in the discussion section add the possibility to use a modified PAI system as described by Nardi et al. in his papers on apical periodontitis. Evaluate PAI in a 5-score scale is really complicated. It can lead to errors of judgment. For future perspective is advisable to use an easier PAI system in a 3-score scale.
In the present study we used for evaluation of the apical periodontitis the PAI in a 5-score scale which is really complicated and it can lead to errors of judgment if the interpretation is made by a beginner practitioner; for future perspective and to avoid classification errors we will use in other studies an easier PAI system in a 3-score scale described by Nardi et al. (2017). [Nardi C, Calistri L, Pradella S, Desideri I, Lorini C, Colagrande S. Accuracy of Orthopantomography for Apical Periodontitis without Endodontic Treatment. J Endod. 2017 Oct;43(10):1640-1646. doi: 10.1016/j.joen.2017.06.020. Epub 2017 Aug 12. PMID: 28807372.]
Conclusions
- Ok
Figures
- Ok
Tables
- Gather together Tables 1-3-5-7
- Gather together Tables 2-4-6-8
Table2 shows the average extent of the periapical lesion recorded at different time intervals; it has a linear regression decreasing from preop to 3, 6 and12 months post-therapy.
Table 2. Mean values and standard deviation of the PAI, at different time intervals, for the group of patients where the dehydrated plant extract, calcium hydroxide, calcium hydroxide with CHX gel 2%, Walkhoff paste were applied.
|
|
Dry vegetable extract |
Ca(OH)2 |
Ca(OH)2 with CHX gel 2% |
Walkhoff paste |
|
Period |
Mean ± SD |
Mean ± SD |
Mean ± SD |
Mean ± SD |
|
Preop |
4±0,73 |
3,9±0,87 |
4,26±0,57 |
4,18±0,47 |
|
3 months |
3,5±1,08 |
3±1,05 |
1,94±1,82 |
1,17±1,33 |
|
6 months |
2,6±1,34 |
1,9±0,99 |
0,32±0,37 |
0,39±0,91 |
|
12 months |
1,6±0,84 |
1,3±0,48 |
0,19±0,42 |
0,03±0,11 |
Ca(OH)2=Calcium hydroxide
SD=Standard deviation
Table 3 shows that there is an average inverse correlation between the value of periapical lesion extent recorded preoperatively and after 12 monthsand also between the two parameters there are differences with high statistical significance (p <0.001).
Table 3. Correlations between PAI values recorded at different time intervals.
|
|
Dry vegetable extract |
Ca(OH)2 |
Ca(OH)2 with CHX gel 2% |
Walkhoff paste |
|
Period |
Correlation index ±p |
Correlation index ± p |
Correlation index ± p |
Correlation index ± p |
|
preop vs. 3 months |
0,88±0,258 |
0,95±0,001 |
0,95±0,001 |
0,93±0,001 |
|
preop vs. 6 months |
0,90±0,011 |
0,84±0,001 |
0,92±0,001 |
0,68±0,001 |
|
preop vs. 12 months |
0,80±0,001 |
0,80±0,001 |
0,58±0,001 |
0,52±0,001 |
|
3 months vs. 6 months |
0,91±0,117 |
0,93±0,021 |
0,96±0,012 |
0,72±0,146 |
|
3 months vs. 12 months |
0,85±0,001 |
0,87±0,001 |
0,74±0,008 |
0,54±0,014 |
|
6 months vs. 12 months |
0,91±0,062 |
0,79±0,148 |
0,95±0,001 |
0,91±0,231 |
Ca(OH)2=Calcium hydroxide
SD=Standard deviation
We are very grateful to you for the review report and for the extremely useful suggestions!
Sincerely,
Dr. Erdogan Elvis Șachir
Reviewer 3 Report
Much effort has been put into this clinical trial but it is poorly presented.
I hope you can benefit from the suggestions bellow.
ABSTRACT
The abstract is not clear. Numbers in brackets should be omitted. The terms used are not common: it is common to use ''microleakage'' instead of ''micro-infiltration''. Although the failure of endodontic treatment is related to apical microleakage, this does not give the complete background of this research. You should mention antibacterial properties of sealers and intracanal medicaments, for example: Sealing abilities and antibacterial properties of root canal sealers and intracanal medicaments between appointments have been recognized as important factors for the success of endodontic treatment. The aim was to evaluate the dynamics of healing by recording PAI, after two appointment endodontic procedure with commercial or experimental intracanal medicament between the appointments. Methods are not clear. Clearly state what was done. How were the patients selected- inclusion and exclusion criteria (based on clinical and radiographic examination)? Which clinical procedures were done? How were the patients followed and PAI recorded? Results: The information whether the difference between the groups (standard vs. experimental with the vegetable oil) is significant or not. Conclusion is not justified by the results. The healing could be attributed to the sealing alone and not to the presence of the vegetable extract.
INTRODUCTION
Concept of the introduction is not good. Why mention anatomy and morphology? This is not your topic.
First describe the objectives of endodontic treatment. Describe how the follow up is done and how the success/failure is evaluated. Define PAI. State which materials are needed in the treatment and justify by references why you chose procedure with more appointments and no single appointment. Which are the characteristics of the materials used for intracanal medicaments between appointments. Then describe the vegetable extract you used and give references supporting your choice of this extract in endodontic treatment. Finally, state the null hypotheses.
Line 41-42
''It has been shown that in the healing of periapical lesions not the quality of canal fillings has a decisive role, but the presence or absence of bacteria in the endodontic space.''
This is not true. In fact, if microleakage takes place due to inadequate filling, the endodontic space gets colonized by bacteria. It sure is important to reduce, or ideally eliminate microorganisms form the endodontic space prior to the root canal filling, but if the quality of the root canal is inadequate it will surely become infected again.
M&M
Line 76 All patients diagnosed with chronic apical periodontitis
Line 90 ''60 patients with failed endodontic treatments or chronic periodontitis…'' Note that chronic periodontitis is another entity. Here the lesions are associated with chronic (peri)apical periodontitis. Moreover, failed endodontic treatments could present themselves as periapical translucencies i.e. chronic apical periodontitis. It would be more precise to say: …'’60 patients with primary chronic apical periodontitis or chronic apical periodontitis due to inadequate endodontic treatment….''
Line 97-103 This paragraph should be removed form M&M section.
Line 104 This paragraph is odd as well. Why do you mention pain? Avoid using ''as we know''
You already stated exclusion and inclusion criteria. It is obvious that anamnesis and clinical and radiologic examination was performed to include/exclude the patients. These paragraphs are redundant. If you feel more information should be given about clinical examination and radiologic assessment, expand the paragraph with inclusion/exclusion criteria.
Line 112 This paragraph should be partly in the INTRODUCTION and partly with inclusion criteria.
Line 122-135 How is this what you state here related to the objective of your study?
RESULTS
Fraphs are Figures. Figure captions should be placed under the figure. Note that all your captions are the same. It would be helpful if the descriptions of the figures were more informative so that it is clear right away to which group it belongs.
DISCUSSION
Start the discussion with the statement whether your null hypotheses were confirmed or rejected and then compare your results with the results from the previous study but also compare your groups. For example, the most favorable outcome was with the Walkhoff paste etc.
CONCLUSIONS are not supported by the results. The healing is a result of the total endodontic procedure, and reduced size of the periapical lesion cannot be attributed solely to the effect of the intracal medicament.
Author Response
The authors acknowledge the useful observations and suggestions of the reviewer’s as concerns the manuscript entitled ”Radioimaging in the evaluation of the therapeutic effect of the vegetable extract obtained from Epilobium parviflorum Schreb”, co-authored by Erdogan Elvis Șachir *, Cristina Gabriela Pușcașu*, Aureliana Caraiane, Gheorghe Raftu, Victoria Badea, Cristina Bartok-Nicolae*, Grierosu Carmen and Ramona Feier.
According to the reviewer’s recommendations, all the suggestions were considered, as follows:
ABSTRACT
The abstract is not clear. Numbers in brackets should be omitted. The terms used are not common: it is common to use ''microleakage'' instead of ''micro-infiltration''. Although the failure of endodontic treatment is related to apical microleakage, this does not give the complete background of this research. You should mention antibacterial properties of sealers and intracanal medicaments, for example: Sealing abilities and antibacterial properties of root canal sealers and intracanal medicaments between appointments have been recognized as important factors for the success of endodontic treatment. The aim was to evaluate the dynamics of healing by recording PAI, after two appointment endodontic procedure with commercial or experimental intracanal medicament between the appointments. Methods are not clear. Clearly state what was done. How were the patients selected- inclusion and exclusion criteria (based on clinical and radiographic examination)? Which clinical procedures were done? How were the patients followed and PAI recorded? Results: The information whether the difference between the groups (standard vs. experimental with the vegetable oil) is significant or not. Conclusion is not justified by the results. The healing could be attributed to the sealing alone and not to the presence of the vegetable extract.
For years, apical microleakage has been considered the main factor in failure in endodontic therapy. Sealing abilities and antibacterial properties of root canal sealers and intracanal medicaments between appointments have been recognized as important factors for the success of endodontic treatment. Background: Apical periodontitis (AP) is an inflammatory disease around the apex of a tooth root.The microorganisms reach the pulp by dentinal tubules especially when there is an open cavity after a coronal fracture and the pulp is in contact with the septic oral environment. The aim was to evaluate the dynamics of healing by recording periapical index (PAI), after two appointment endodontic procedure with commercial or experimentalintracanal medicament between the appointments.Methods: A total of 40 patients with primary chronic apical periodontitis requiring root canal treatment were assigned randomly into four groups according to the teeth medicated with dehydrated plant extract, calcium hydroxide, calcium hydroxide mixed with chlorhexidine(CHX) gel 2%, Walkhoff pasteand obturated in a second visit 7 days later. Patients were recalled at intervals of 3, 6, and 12 months toevaluate the treated teeth both clinically and radiographically for periapical healing. A 5-score scale PAI was used to evaluate stages of the periapical healing on a periapical radiography using a Kodak Dental imaging software provided by the radio-imagistic center. Results: Radiological evaluation revealed that the experimental intracanal medicamenthad a cumulative positive healing capacity by reducing the PAIas well as all resorbable pastes used in endodontic conventional therapy.Conclusions: The results suggest that the vegetable dry extract obtained from Epilobium parviflorum Schrebcan be used as an inter-appointment medication among with the root canal filling for theapical healing quantified by reducing the PAI.
INTRODUCTION
Concept of the introduction is not good. Why mention anatomy and morphology? This is not your topic.
First describe the objectives of endodontic treatment. Describe how the follow up is done and how the success/failure is evaluated. Define PAI. State which materials are needed in the treatment and justify by references why you chose procedure with more appointments and no single appointment. Which are the characteristics of the materials used for intracanal medicaments between appointments. Then describe the vegetable extract you used and give references supporting your choice of this extract in endodontic treatment. Finally, state the null hypotheses.
The infection of the dental root canal system is called an endodontic infection. This infection can cause an apical periodontitis and the progression of different forms of apical periodontitis is due to some microorganisms [1,2]. The microorganisms reach the pulp by dentinal tubules especially when there is an open cavity after a coronal fracture and the pulp is in contact with the septic oral environment. Another pathway for the microorganism is the periodontal membrane when they use the lateral channel or the apical foramen [1,3]. Epidemiological surveys have shown apical periodontitis to be frequently associated with root-filled teeth. Despite technological advances made during the last couple of decades, studies continue to find high frequencies of sub-standard root fillings. Some reports show improvement in the quality of root fillings as assessed radiographically, but without a concomitant improvement in the apical health in connection with the root-filled teeth [2,3]. Various studies have elucidated the importance of quality of root canal filling and coronal seal in the success for non-surgical endodontic therapy [4]
The imagistic diagnostic techniques are essential from the diagnostic in various domains of the dental medicine. As the aim of the endodontic treatment is to maintain healthy periapical status, the radiographic exam allows the assessment of the changes of the periapical area related to the bone density and the progression of the periapical inflammation. The absence of the periapical radio transparence confirms the effectiveness of the cleaning, disinfection and sealing provided by the root filling and coronal filling [5]. The conventional radiography provides to the practitioner an acceptable, effective and cheap method. The limitation of conventional radiographic exam in-creased the interest for cone beam computed tomography (CBCT), Despite the limited use of only 17% investigations in endodontics field. However, CBCT should not be used routinely in the diagnosis of periapical lesions and endodontic applications due to the ALARA (As Low As Reasonably Achievable) principles [6].
Radiographic evaluation of the apical periodontitis can be made using the PAI scoring system given by Orstavik in 1986. This is a 5-point scale radiographic interpretation designed to determine the absence, presence, or transformation of a diseased state. The reference is made up of a set of five radiographs with corresponding line drawings and their associated score on a photographic print. Table 1 represents the description of PAI scores [7].
Table 1. Description of Periapical index scores (adapted from Orstavik et al. [7])
PAI Score Description of Radiographic findings
1 Normal Periapical Structures
2 Small changes in Bone Structures
3 Change in Bone Structure with Mineral Loss
4 Periodontitis with well-defined radiolucent area
5 Severe periodontitis with exacerbating features
Endodontic materials are used in endodontic treatment, which is the procedure to save the tooth when the pulp and/or peri radicular tissues are injured. These materials can be generally classified into two groups: materials used to maintain the vitality of the pulp (pulp capping materials), and materials used to disinfect (irrigants and intra-canal medicaments) and fill the pulp in root canal therapy [8].
The use of an interappointment medicament has been shown to significantly im-prove disinfection after chemo mechanical procedures. Calcium hydroxide is a commonly used intracanal medicament. It possesses several advantages such as tissue dis-solving ability and antibacterial properties [9]. Chlorhexidine (CHX) is an antiseptic with a broad antimicrobial spectrum and high substantivity. It is commonly used as interappointment root canal medication [10]. Camphor mono-chlorophenol (CMCP) have high antimicrobial activity. However, the use of formocresol which is the most common used material and also the golden standard for medicament is controversial due to its potential cytotoxicity. It has been associated with carcinogenicity, immuno-logical changes, cytotoxicity, teratogenicity, mutagenic effects and causing enamel defects in permanent teeth and systemic changes in internal organs such as the kidneys and the liver. CMCP is also a phenolic derivative that can stimulate periapical tissues at higher concentrations. Therefore, an alternative material with high efficacies would be of utmost clinical interest [11].
Single-visit root canal treatment attempts instrumentation, disinfection and obturation of the root canal system in one visit. In contrast, multiple-visit root canal treatment performs the instrumentation (or large parts of it) in the first and the obturation in the second visit, while the disinfection is provided in both visits via irrigation. Moreover, a disinfecting medication is placed in the canals between visits to allow further reduction of bacterial numbers. While single-visit treatment has obvious ad-vantages over conventional multiple-visit treatment (like reduced number of visits, no need for repeated application of anesthetics or rubber dam, no intermediary restoration); it might be disadvantageous both with regard to short-term and long-term out-comes [12].
Recently, the focus of research has shifted towards finding natural alternatives to synthetic medications. Several natural products have been tested as intracanal medicaments for this purpose, namely: Liquorice extract, propolis, Morindacontroflia, and the extract of Arctium lappa. Propolis is a natural, biocompatible, and resinous byproduct, which has been used extensively as an organic medicine for managing oral and throat infection, dental caries, skin wounds, burns, leg ulcers, psoriasis, atopic dermatitis, recurrent aphthous ulcers, warts, herpes labialis, herpes genitalis, wound healing, tissue regeneration, and as a diet supplement. Moreover, studies suggest that it can be used to prevent and treat the cases of oral mucositis induced due to cancer therapy and to control diabetes and obesity complications. It is prepared from sub-stances collected from different parts of plants and deposited into a beehive by honey bees (Apis mellifera L.). It possesses anti-microbial and anti-inflammatory properties due to the presence of flavonoids. Moreover, the anti-microbial properties of propolis have been proven to be superior to calcium hydroxide in vitro. Similarly, two clinical studies have found propolis to be superior to calcium hydroxide when used as a direct pulp capping agent and as an intracanal irrigant [13].
The anti-inflammatory and antiproliferative activity of Epilobium parviflorum Schreb extracts was initially studied on cells in the context of benign prostatic hyperplasia, in parallel the analgesic, antioxidant, antibacterial and antifungal action were studied [14-16]. Plant polyphenols represent a group of chemical substances ubiquitously distributed in all higher plants. These secondary metabolites possess free radical scavenging and antimicrobial activity. These properties can be advantageously ex-ploited, especially because of the abundance of polyphenols and their derivatives in various agricultural and food industry waste and by‐products and the possibility of convenient ex-traction by either organic or aqueous solvents [17-19].
No studies have assessed the efficacy of Epilobium parviflorum Schreb as a “medicament” for root canal treatment of apical periodontitis, therefore this study aimed to evaluate the dynamics of healing by recording PAI, after two appointment endodontic procedure with commercial or experimental intracanal medicament between the appointments.
Line 41-42
''It has been shown that in the healing of periapical lesions not the quality of canal fillings has a decisive role, but the presence or absence of bacteria in the endodontic space.''
This is not true. In fact, if microleakage takes place due to inadequate filling, the endodontic space gets colonized by bacteria. It sure is important to reduce, or ideally eliminate microorganisms form the endodontic space prior to the root canal filling, but if the quality of the root canal is inadequate it will surely become infected again.
We have removed the sentence.
M&M
Line 76 All patients diagnosed with chronic apical periodontitis
We have changed with apical periodontitis
Line 90 ''60 patients with failed endodontic treatments or chronic periodontitis…'' Note that chronic periodontitis is another entity. Here the lesions are associated with chronic (peri)apical periodontitis. Moreover, failed endodontic treatments could present themselves as periapical translucencies i.e. chronic apical periodontitis. It would be more precise to say: …'’60 patients with primary chronic apical periodontitis or chronic apical periodontitis due to inadequate endodontic treatment….''
We have changed with60 patients with primary chronic apical periodontitis or chronic apical periodontitis due to inadequate endodontic treatment
Line 97-103 This paragraph should be removed form M&M section.
We have removed it.
Line 104 This paragraph is odd as well. Why do you mention pain? Avoid using ''as we know''
You already stated exclusion and inclusion criteria. It is obvious that anamnesis and clinical and radiologic examination was performed to include/exclude the patients. These paragraphs are redundant. If you feel more information should be given about clinical examination and radiologic assessment, expand the paragraph with inclusion/exclusion criteria.
We have removed the paragraph ”Pain is one of the main symptoms that occur in endodontic diseases, but as we know, dental and facial pain can be caused by various causes, it is important to establish a correct diagnosis.” Line 104
Line 112 This paragraph should be partly in the INTRODUCTION and partly with inclusion criteria.
We have removed the 2.4 Recording the PAI score, it was placed in introduction.
Line 122-135 How is this what you state here related to the objective of your study?
We have removed the paragraph between line 122-135 we have intitled the properties of an endodontic materials needed in the treatment.
RESULTS
Fraphs are Figures. Figure captions should be placed under the figure. Note that all your captions are the same. It would be helpful if the descriptions of the figures were more informative so that it is clear right away to which group it belongs.
We have changed the captions for each graph so figures could be more informative by offering to the viewer a clear group belonging.
DISCUSSION
Start the discussion with the statement whether your null hypotheses were confirmed or rejected and then compare your results with the results from the previous study but also compare your groups. For example, the most favorable outcome was with the Walkhoff paste etc.
The limitations of the study are given by the final number of the study group and the 5-score scale PAI which is really complicated and it can lead to errors of judgment if the interpretation is made by a beginner practitioner, in order to overcame these limitations for future perspective, it is necessary to conduct a study with a larger number of patients and to avoid classification errors we will use in other studies an easier PAI system in a 3-score scale described by Nardi et al. (2017) [24].
In the present study the paste obtained from the dehydrated vegetable extract had a therapeutic effect, managing to produce apical healing in a percentage of 60%, reducing the PAI score from 4 preoperative to 1,6 after 12 months recall. The data obtained from the statistical processing demonstrate, that the therapeutic value of the experimental paste is above to that of the commercial medicament used in endodontic practice. In the accessed literature we did not find studies to test the therapeutic effect of alcoholic, aqueous or dehydrated vegetable extract obtained from Epilobium parviflorum Schreb in endodontic therapy.
The most favorable outcome was with the Walkhoff paste which produced apical healing in a percentage of 90%, reducing the PAI score from 4,18 preoperative to 0,03 after 12 months recall; although difficult to compare due to the methods and techniques used, the data obtained in this study are consistent with those of the study by Zhila Imani et al (2018), who demonstrated by bacteriological techniques the therapeutic efficacy of the p-monochlorophenol solution in Walkhoff paste a high capacity for apical healing [11].
The group with calcium hydroxide had an apical healing in a proportion of 70%, reducing the PAI score from 3,9 preoperative to 1,3 after 12 months recall. The results are similar to the data obtained by Sahara et al. (2019) which using the same method of quantifying apical healing, PAI decreased from scale 5 preoperative to 2 after 3 months recall [25]. In previously published studies we found data that show that the therapeutic effect of calcium hydroxide is time dependent as it is in the study con-ducted by Gheorghiu Maria et al. (2018). They have shown that calcium hydroxide has a very high efficiency when placed in the root canals for a longer period of time (10 days) and extremely low efficiency after 48 hours [26].
The group with calcium hydroxide paste mixed with CHX 2% had a therapeutic effect on apical healing in a percentage of 80%, reducing the PAI score from 4,26 pre-operative to 0,19 after 12 months recall. The data obtained in the present study are close to the results obtained by Ertugrul Ercan et al. (2007), who used the whole PAI at the same time intervals to evaluate periapical healing. The results obtained were 64.1%, although the CHX concentration was 1%; the difference in percentage between the two studies can be determined both by the concentration of CHX used and by its forms of presentation (liquid/gel) [27].
The results of the present study revealed significant information on intracanal medications, commercial and experimental, role in the apical healing process monitored by recording the PAI score at 3, 6 and 12 months recall after root canal filling on a periapical radiography.
CONCLUSIONS are not supported by the results. The healing is a result of the total endodontic procedure, and reduced size of the periapical lesion cannot be attributed solely to the effect of the intracal medicament.
The results suggest that the vegetable dry extract obtained from Epilobium parviflorum Schrebcan be used as an inter-appointment medication among with the root canal filling for the apical healing quantified by reducing the PAI.
We are very grateful to you for the review report and for the extremely useful suggestions!
Sincerely,
Dr. Erdogan Elvis Șachir
Round 2
Reviewer 1 Report
The manuscript may require assistance in English grammar.
The remarks have been given in the attachment document. The author has done good work but has not been very scientific when writing the manuscript.

Author Response
The authors acknowledge the useful observations and suggestions of the reviewer’s as concerns the manuscript entitled ”Radioimaging in the evaluation of the therapeutic effect of the vegetable extract obtained from Epilobium parviflorum Schreb”, co-authored by Erdogan Elvis Șachir *, Cristina Gabriela Pușcașu*, Aureliana Caraiane, Gheorghe Raftu, Victoria Badea, Cristina Bartok-Nicolae*, Grierosu Carmen and Ramona Feier.
According to the reviewer’s recommendations, all the suggestions were considered, as follows:
1.This infection can cause an apical periodontitis and the progression of different forms of apical periodontitis is due to some microorganisms [1,2].
The infection of the dental root canal system is called an endodontic infection which can cause an apical periodontitis, defined as a dynamic encounterbetween the microbes and host defense system at theinterface between the infected radicular pulp and periodontalligament. This results in inflammation, periapicaldestruction and resorption, eventually manifesting asvarious histopathological forms of periapical lesions[1,2].
2.As the aim of the endodontic treatment is to maintain healthy periapical status, the radiographic exam allows the assessment of the changes of the periapical area related to the bone density and the progression of the periapical inflammation.
As the aim of the endodontic treatment is to maintain healthy periapical status, the radiographic examination allows the assessment of the changes ofthe periapical area related to the bone density and the progression of the periapical inflammation[6].
- The limitation of conventional radiographic exam increased the interest for cone beam computed tomography (CBCT), Despite the limited use of only 17% investigations in endodontics field.
Ideal features of an imaging system are geometric accuracy,minimal superimposition, ease of availability andusage, reliable, reproducible, relatively inexpensive, andmost importantly minimum radiation exposure to thepatient [8]. Superimpositionand image distortion can be some potentialdisadvantagesof conventional radiographic examinationwhich increased the interest for cone beam computed tomography (CBCT), despite the limited use of only 17% investigations in endodontics field. However, CBCT should not be used routinely in the diagnosis of periapical lesions and endodontic applications due to the ALARA (As Low AsReasonably Achievable) principles[6,8].
- However, the use of formocresol which is the most common used material and also the golden standard for medicament
We have dropped on the sentence referring the formocresol as golden standard foe medicament.
- Single-visit root canal treatment attempts instrumentation, disinfection and obturation of the root canal system in one visit. In contrast, multiple-visit root canal treatment performs the instrumentation (or large parts of it) in the first and the obturation in the second visit, while the disinfection is provided in both visits via irrigation. Moreover, a disinfecting medication is placed in the canals between visits to allow further reduction of bacterial numbers. While single-visit treatment has obvious advantages over conventional multiple-visit treatment (like reduced number of visits, no need for repeated application of anesthetics or rubber dam, no intermediary restoration); it might be disadvantageous both with regard to short-term and long-term outcomes [12].
We have dropped on the entire paragraph.
6.No studies have assessed the efficacy of Epilobium parviflorum Schreb as a “medicament” for root canal treatment of apical periodontitis, therefore this study aimed to evaluate the dynamics of healing by recording PAI, after two appointment endodontic procedure with commercial or experimental intracanal medicament between the appointments.
There arestudies (V. Steenkampet al., 2006, Bajer T. et al., 2017) regarding the antibacterial activityof Epilobium parviflorum Schrebspecies but none of themassessed the efficacy of Epilobium parviflorum Schreb as a “medicament” for root canal treatment in apical periodontitis; therefore this study aimed to evaluate the dynamics of healing by recording PAI, after two appointment endodontic procedure with commercial or experimental intracanal medicament between the appointments [21, 22].
- All patients diagnosed with apical periodontitis
All patients with posterior teeth diagnosed with pulp necrosis, (i.e. negative response to pulp sensitivity test with cold stimulus, confirmed by absence of bleeding during access cavity preparation) asymptomatic or symptomatic apical periodontitis.
- incorrect endodontic treatment
The inclusion criteria in the study were: participants had to be over 18 years of agewithout allergic reactions with posterior teeth diagnosed with pulp necrosis. Moreover, the included teeth also had either symptomatic apical periodontitis, asymptomatic apical periodontitis, or chronic apical abscessassociated with a periapical radiolucency.
- primary chronic apical periodontitis or chronic apical periodontitis due to inadequate endodontic treatment
asymptomatic or symptomatic apical periodontitis
- After accessing the tooth, the canals were prepared bypreflaring the coronal (or straight) portion prior to negotiation of the apical portion and determination of working length.
Root treatment procedures were completed under the same protocols by an experienced endodontist. All treatments were performed under topic and local anesthesia (Lidocaine spray 10%, Mepivastesin 3%/Ubistesin 4% 3M ESPE Germany).Single tooth isolation was undertaken using a rubber dam and light-cured gingival barrier (Dam Liquid, Cerkamed, Stalowa Wola, Poland). The surface of the rubber dam and the tooth were disinfected by swabbing for 60 seconds with rubbing alcohol 70% (ethanol-based liquid, Alcomar, Romania).Coronal access was achieved with a sterile bur followed by canals initial scouting with size10 or size 15 Kerr file (Dentsply Sirona). Working length was determined using an electronic apex locator (Denjoy Joypex 5, Skysea) and confirmed with a digital periapical radiograph.Canal preparation was performed in a crown-down approach using rotary instruments (ProTaper® Gold; Dentsply Sirona) at 300 RPM and a maximum torque of 4N to a size F2 master apical size, whenever possible.
For the teeth with previously root canal treatment, former root canal obturation material was mechanically removed using ProTaper® RT files (Dentsply Maillefer, Ballaigues,Switzerland) coded D1 to D3. After reaching the original apical foramen using hand filesfrom #6-15; working length was determined and the root instrumentation performed asthe above-described technique.
On the commercial medicament groups the canals were disinfected with 2% sodium hypochlorite (NaOCl) throughout the procedure, delivered using a side-vented 30-gauge needle(Mekan Med., Shanghai, China).TheNaOCl activations were made by ultrasonic passive technique using the protocol describedby Pirela et al. (2020) [24]. Final irrigation was made with 2% NaOCl followed by sterile saline solution, and 17% ethylenediaminetetraacetic acid (EDTA, Endo-solution, Cerkamed, Stalowa Wola, Poland). After a final rinse with saline, the root canals were dried with sterile paper points (Absorbent Paper Points, Metabiomed).Only the NaOCl was entirely replaced with the hydroalcoholic plant extract 25% on the group with theexperimental medicament, the rest of the irrigation protocol remain unchanged.
An interappointment dressing of experimental or commercial paste was placed repetitively with a Kerr file 20 ISO by counterclockwise rotational movements associated with vertical plunging movements until the intracanal medication suppress in the cavityaccess; a temporarily dressed glass ionomer cement (GC Fuji, Minnesota, USA) was used [25].
- After 1 week, the root canals were reentered and the temporary antiseptic paste from the root canal was removed using a Kerr file 25 ISO or a ProTaper Universal F1.
After 1 week, the root canals were reentered; in the groups with commercial medicament the temporary antiseptic paste from the root canal was removed using a copious irrigation withNaOCl 2%, EDTA 17%, citric acid40% (Cerkamed,Stalowa Wola, Poland)in combination with hand filing (Kerr file 25 ISO). For more efficient elimination of intracanal medicament, passive ultrasonics were used according with the protocol described by Raghu et al (2017). This is accomplished with a small file (20 or 25 ISO) vibrated in a previously shaped root canal to produce acoustic streaming that transfers its energy to the irrigant inside the canal.In the group with the experimental medicament the above protocol of removal temporary paste was used, only the NaOCl was entirely replaced with the hydroalcoholic plant extract 25%[26].
- The patients were sent to the same radiological center to perform an immediate post-operative radiograph control using a periapical radiography as well as at 3, 6, 12 monthsrecall.
Radiological evaluation was performed on 3 months after endodontic treatment in accordance with other previous studies (Best S. et al., 2014, Salceanu M. et al., 2021) and with the Romanian Endodontics Guide (Andrei Iliescu, 2015) which establishes the appearance of radiologically visible apical healing signs at 3 months after performing the root canal treatment [30-32].
- 12 months after endodontic treatment
After root filling teeth werereferredto a prosthetic specialist for a crown coverage if indicated. During the survey,temporary prosthetic crown according to Heimann (2021) was made by a modified radiolucent hybrid composite (Estenia, Kuraray) to avoid interference withthe radiological assessment and diagnosis ofmarginal gaps, quality of the root filling and the overall condition forapical healing [29].
- Discussion
The objective in this study wasto assess radiographic healing of teeth treated with commercial or experimental intracanal medicament between twoendodontic appointments.
To evaluate radiographic healing patterns, it was used the PAI system developed by Orstavik et al (1986). Since its inception, this scoring system has become increasingly popular in endodontic outcome studies [9]. At baseline, the teeth were classified as having a radiolucent lesion with a PAI score mean>4. By the latest follow-up, all of the teeth had a PAI score mean of <1.3;regression of the PAI score mean forall four substances studied demonstrates the effectiveness of the intracanal medicaments.
The ratio of cured teeth is majority in each study group, which demonstrates the effectiveness of the substances applied in endodontic therapy among with the canal filling and the success of endodontic treatment by radiological monitoring of apical healing.
The results of the present study revealed significant information on intracanal medications, commercial and experimental, role in the apical healing process monitored by recording the PAI score at 3, 6 and 12 months recall after root canal filling on a periapical radiography.
In the present study the dehydrated vegetable extract had a therapeutic effect, managing to produce apical healing in a percentage of 60%, reducing the PAI score from 4 preoperative to 1,6 after 12 months recall. The data obtained from the statistical processing demonstrate, that the therapeutic value of the experimental paste is above to that of the commercial medicament used in endodontic practice. In the accessed literature we did not find studies to test the therapeutic effect of alcoholic, aqueous or dehydrated vegetable extract obtained from Epilobium parviflorum Schreb in endodontic therapy.
The most favorable outcome was with the Walkhoff paste which produced apical healing in a percentage of 90%, reducing the PAI score from 4,18 preoperative to 0,03 after 12 months recall; although difficult to compare due to the methods and techniques used, the data obtained in this study are consistent with those of the study by Zhila Imani et al (2018), who demonstrated by bacteriological techniques the therapeutic efficacy of the p-monochlorophenol solution in Walkhoff paste a high capacity for apical healing [13].
The group with calcium hydroxide had an apical healing in a proportion of 70%, reducing the PAI score from 3,9 preoperative to 1,3 after 12 months recall. The results are similar to the data obtained by Sahara et al. (2019) which using the same method of quantifying apical healing, PAI decreased from scale 5 preoperative to 2 after 3 months recall [33]. In previously published studies we found data that show that the therapeutic effect of calcium hydroxide is time dependent as it is in the study con-ducted by Gheorghiu Maria et al. (2018). They have shown that calcium hydroxide has a very high efficiency when placed in the root canals for a longer period of time (10 days) and extremely low efficiency after 48 hours [34].
The group with calcium hydroxide paste mixed with CHX 2% had a therapeutic effect on apical healing in a percentage of 80%, reducing the PAI score from 4,26 pre-operative to 0,19 after 12 months recall. The data obtained in the present study are close to the results obtained by Ertugrul Ercan et al. (2007), who used the whole PAI at the same time intervals to evaluate periapical healing. The results obtained were 64.1%, although the CHX concentration was 1%; the difference in percentage between the two studies can be determined both by the concentration of CHX used and by its forms of presentation (liquid/gel) [35].
The limitations of the study are given by the final number of the study group and the 5-score scale PAI which is really complicated and it can lead to errors of judgment if the interpretation is made by a beginner practitioner, in order to overcame these limitations for future perspective, it is necessary to conduct a study with a larger number of patients and to avoid classification errors we will use in other studies an easier PAI system in a 3-score scale described by Nardi et al. (2017) [36].
We are very grateful to you for the review report and for the extremely useful suggestions!
Sincerely,
Dr. Erdogan Elvis Șachir
Reviewer 2 Report
I really appreciated the efforts made by the authors.
Author Response
The authors acknowledge the useful observations and suggestions of the reviewer’s as concerns the manuscript entitled Radioimaging in the evaluation of the therapeutic effect of the vegetable extract obtained from Epilobium parviflorum Schreb, co-authored by Erdogan Elvis Șachir *, Cristina Gabriela Pușcașu*, Aureliana Caraiane, Gheorghe Raftu, Victoria Badea, Cristina Bartok-Nicolae*, Grierosu Carmen and Ramona Feier.
According to the reviewer’s recommendations, all the suggestions were considered, as follows:
English language and style are fine/minor spell check required
We have improved the English language grammar.
We are very grateful to you for the review report and for the extremely useful suggestions!
Sincerely,
Dr. Erdogan Elvis Șachir
Reviewer 3 Report
The article has been considerably improved. I suggest placing the limitations of the study at the end of the discussion instead.
Author Response
The authors acknowledge the useful observations and suggestions of the reviewer’s as concerns the manuscript entitled ”Radioimaging in the evaluation of the therapeutic effect of the vegetable extract obtained from Epilobium parviflorum Schreb”, co-authored by Erdogan Elvis Șachir *, Cristina Gabriela Pușcașu*, Aureliana Caraiane, Gheorghe Raftu, Victoria Badea, Cristina Bartok-Nicolae*, Grierosu Carmen and Ramona Feier.
According to the reviewer’s recommendations, all the suggestions were considered, as follows:
English language and style are fine/minor spell check required
We have improved the English language grammar
I suggest placing the limitations of the study at the end of the discussion instead.
We have placed the limitations of the study at the end of the discussion.
The objective in this study was to assess radiographic healing of teeth treated with commercial or experimental intracanal medicament between two endodontic appointments.
To evaluate radiographic healing patterns, it was used the PAI system developed by Orstavik et al (1986). Since its inception, this scoring system has become increasingly popular in endodontic outcome studies [9]. At baseline, the teeth were classified as having a radiolucent lesion with a PAI score mean >4. By the latest follow-up, all of the teeth had a PAI score mean of <1.3; regression of the PAI score mean for all four substances studied demonstrates the effectiveness of the intracanal medicaments.
The ratio of cured teeth is majority in each study group, which demonstrates the effectiveness of the substances applied in endodontic therapy among with the canal filling and the success of endodontic treatment by radiological monitoring of apical healing.
The results of the present study revealed significant information on intracanal medications, commercial and experimental, role in the apical healing process monitored by recording the PAI score at 3, 6 and 12 months recall after root canal filling on a periapical radiography.
In the present study the dehydrated vegetable extract had a therapeutic effect, managing to produce apical healing in a percentage of 60%, reducing the PAI score from 4 preoperative to 1,6 after 12 months recall. The data obtained from the statistical processing demonstrate, that the therapeutic value of the experimental paste is above to that of the commercial medicament used in endodontic practice. In the accessed literature we did not find studies to test the therapeutic effect of alcoholic, aqueous or dehydrated vegetable extract obtained from Epilobium parviflorum Schreb in endodontic therapy.
The most favorable outcome was with the Walkhoff paste which produced apical healing in a percentage of 90%, reducing the PAI score from 4,18 preoperative to 0,03 after 12 months recall; although difficult to compare due to the methods and techniques used, the data obtained in this study are consistent with those of the study by Zhila Imani et al (2018), who demonstrated by bacteriological techniques the therapeutic efficacy of the p-monochlorophenol solution in Walkhoff paste a high capacity for apical healing [13].
The group with calcium hydroxide had an apical healing in a proportion of 70%, reducing the PAI score from 3,9 preoperative to 1,3 after 12 months recall. The results are similar to the data obtained by Sahara et al. (2019) which using the same method of quantifying apical healing, PAI decreased from scale 5 preoperative to 2 after 3 months recall [33]. In previously published studies we found data that show that the therapeutic effect of calcium hydroxide is time dependent as it is in the study con-ducted by Gheorghiu Maria et al. (2018). They have shown that calcium hydroxide has a very high efficiency when placed in the root canals for a longer period of time (10 days) and extremely low efficiency after 48 hours [34].
The group with calcium hydroxide paste mixed with CHX 2% had a therapeutic effect on apical healing in a percentage of 80%, reducing the PAI score from 4,26 pre-operative to 0,19 after 12 months recall. The data obtained in the present study are close to the results obtained by Ertugrul Ercan et al. (2007), who used the whole PAI at the same time intervals to evaluate periapical healing. The results obtained were 64.1%, although the CHX concentration was 1%; the difference in percentage between the two studies can be determined both by the concentration of CHX used and by its forms of presentation (liquid/gel) [35].
The limitations of the study are given by the final number of the study group and the 5-score scale PAI which is really complicated and it can lead to errors of judgment if the interpretation is made by a beginner practitioner, in order to overcame these limitations for future perspective, it is necessary to conduct a study with a larger number of patients and to avoid classification errors we will use in other studies an easier PAI system in a 3-score scale described by Nardi et al. (2017) [36].
We are very grateful to you for the review report and for the extremely useful suggestions!
Sincerely,
Dr. Erdogan Elvis Șachir